# OSCAR: ONLINE SOFT COMPRESSION FOR RAG

**Maxime Louis***
Naver Labs Europe
maxime.louis.x2012@gmail.com

**Thibault Formal**
Naver Labs Europe
thibault.formal@gmail.com

**Hervé Déjean**
Naver Labs Europe
herve.dejean@naverlabs.com

**Stéphane Clinchant**
Naver Labs Europe
stephane.clinchant@naverlabs.com

## ABSTRACT

Retrieval-Augmented Generation (RAG) enhances large language models (LLMs) by integrating external knowledge, leading to improved accuracy and relevance. However, scaling RAG pipelines remains computationally expensive as context length grows. To address this, hard compression methods prune the retrieved text on-the-fly, achieving only modest compression ratios, whereas soft compression methods rely on costly offline LLM-based compression to obtain higher rates. In this paper, we introduce OSCAR, a novel query-dependent online soft compression method for RAG. OSCAR bridges the gap between online hard and offline soft compression methods, bringing the best of both: OSCAR dynamically compresses retrieved documents into a representation optimized for the query at hand, leading to efficient and accurate downstream answer generation. Our experiments demonstrate state-of-the-art performance with a 2–5× speed-up in inference and minimal, if any, accuracy loss, for LLMs ranging from 1B to 24B parameters.

## 1 INTRODUCTION

Retrieval-Augmented Generation (RAG) (Lewis et al., 2020; Guu et al., 2020; Borgeaud et al., 2022) has become pivotal for solving a wide range of natural language processing challenges. RAG enhances Large Language Models (LLMs) by leveraging retrieved documents from curated datasets, enabling more accurate, well-grounded, and up-to-date responses. However, one major issue when scaling up RAG pipelines is the high computational cost.

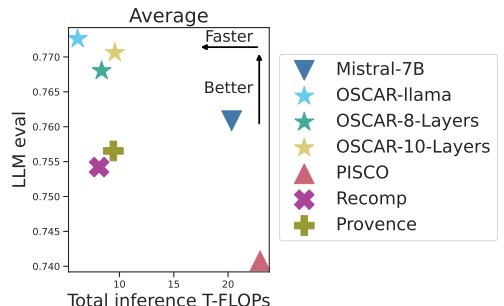

Figure 1: OSCAR models enable faster end-to-end inference with retrieval as well as improved accuracy compared to hard compression methods.

To improve efficiency, a natural idea consists in replacing the retrieved documents with a more compact representation. A straightforward option is to perform *hard* compression on the text itself to form a summarized or pruned version as in Xu et al. (2023); Kenton & Toutanova (2019); Wang et al. (2023). These methods are LLM-agnostic and robust, but their compression rates are modest ($\simeq$ ×2), limiting overall efficiency gains. Most hard compression methods operate in an online, query-aware fashion, dynamically compressing the documents to maximize utility for the task.

Another option is *soft* compression which maps retrieved texts to a continuous embedding space. Typically, texts are mapped to a K/V cache (Qin et al., 2024) or to an embedding which can be fed into the transformer by bypassing its embedding layer (Chevalier et al., 2023; Ge et al., 2023; Hofstätter et al., 2023; Louis et al., 2025; Rau et al., 2024b). These approaches achieve higher compression ($\simeq$ ×16), but at the cost of substantial performance degradation, and they fall short of

---

*Corresponding author.

empirical and theoretical efficiency bounds (Kuratov et al., 2025). In fact, we note that none of the existing soft compression methods use the query at compression time: all of them rely on heavy, LLM-sized forward passes performed offline, as doing it online would not lead to overall efficiency improvements.

Both hard and soft compression thus have complementary strengths: hard methods are online and query-aware but limited in compression rate, while soft methods promise higher rates but suffer from quality loss are not usable online. Ideally, one would combine the advantages of both—high compression with query-dependent online operation. However, designing a fast enough compression operator remains an open challenge: existing methods either sacrifice efficiency, accuracy or fail to scale to dynamic RAG scenarios. Developing an efficient online compression strategy would also facilitate dynamic RAG scenarios in which retrieved content originates from the open web or from large-scale corpora in a plug-and-play manner.

In this paper, we show how to build an efficient compression model to obtain large efficiency improvements in RAG pipelines. **The obtained OSCAR models—for Online Soft Compression for RAG—are novel soft-compression query-dependent methods for RAG. We obtain 2-5x faster end-to-end inference on a variety of LLMs ranging from 1B to 24B parameters[1]. Crucially, the obtained models suffer from little to no accuracy loss on a variety of in-domain and out-of-domain RAG benchmarks.** Lastly, we notice, as discussed by Chirkova et al. (2025), that the compression operation can be exploited to simultaneously re-rank the initial pool of retrieved documents. Since re-ranking is an integral part of efficient RAG pipelines (Rau et al., 2024a), this enables us to obtain the compression representation of the documents for free.

## 2 RELATED WORKS

**Long context optimizations for RAG** RAG scaling problems relate to the long-context (in)abilities of LLMs which is an active area of research. K/V caching techniques enable faster long context handling by diminishing the number of operations in self-attention (Devoto et al., 2024; Kwon et al., 2023; Li et al., 2024). FINCH (Corallo & Papotti, 2024) is more specifically designed for RAG: the retrieved content is chunked and only a small portion of the keys and values is kept in cache for each chunk for the subsequent attention computations – but compression rates remain limited. TurboRAG and block-attention RAG (Sun et al., 2024; Lu et al., 2024) propose to modify the attention causal mask to compute attention independently on each retrieved documents, while the query still attends to each previous token in the context. Overall, we note that these KV-cache compression methods are orthogonal to our work: combining both would be possible.

**Hard compression methods** aim at shortening the retrieved documents by summarization or pruning. Most of them have limited compression rates due to the nature of text but are agnostic to the LLM used for generation. Provence (Chirkova et al., 2025) proposes to fine-tune a DeBERTa (He et al., 2021) model to prune retrieved contexts. It is fast, prunes the context in a query-dependent fashion and allows the simultaneous reranking of the retrieved documents—making pruning essentially free in a standard RAG pipeline. Extractive RECOMP (Xu et al., 2023) prunes contexts based on sentences embeddings. Abstractive RECOMP summarizes input contexts using an autoregressive LLM: the efficiency improvement is less clear than Provence since generating the summary is an expensive operation. Other methods include FILCO (Wang et al., 2023) or COMPACT (Yoon et al., 2024), which also generate pruned contexts autoregressively.

**Soft compression methods** aim at compressing retrieved documents into vector representations, often to be used as input embeddings or K/V cache to the LLM used for generation. These methods generally achieve higher compression rates but require a training specific to the LLM used for generation. xRAG (Cheng et al., 2024) proposes to use retrieval embeddings as precomputed compressed representations, and trains an adapter MLP to map these embeddings into inputs for the LLM – performances remain however limited. COCOM (Rau et al., 2024b), building on (Chevalier et al., 2023; Ge et al., 2023), proposes an end-to-end training pipeline where both the compression LLM and the generation LLM are fine-tuned using a large QA dataset. PISCO (Louis et al., 2025) is an extension of COCOM trained by sentence-level distillation from a teacher LLM: it allows to

---

[1]We release open-source models on 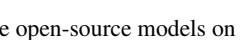 naver/oscar, as well as our training code: github.com/naver/pisco

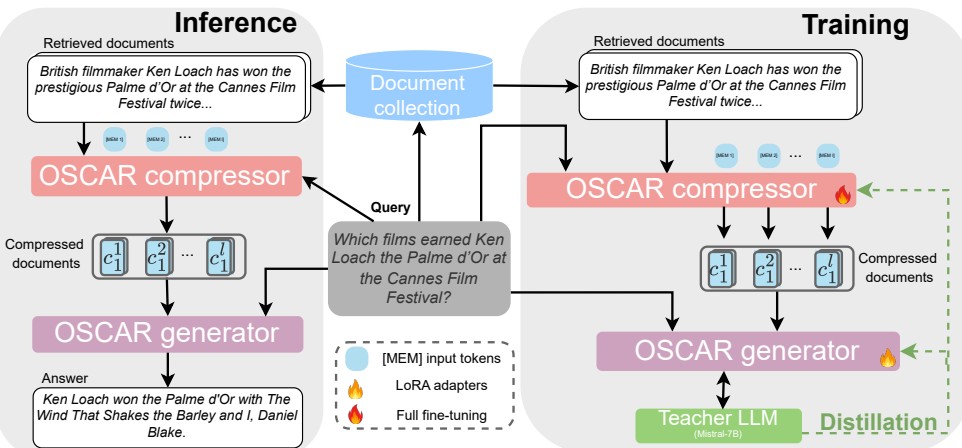

Figure 2: **OSCAR overview.**

compress contexts by a factor of 16× with very limited performance drops. All these approaches process documents independently from the query – attempting to compress all the information of the retrieved documents into the (compressed) vector representation. FiD-light (Hofstätter et al., 2023) proposes a form of query-dependent soft compression by using an encoder-decoder LLM, where the encoder is fed in parallel with the input query and each retrieved document. FiD-light decoder then takes only the first 50 hidden states for each document and thus has a very limited compression rate. Finally, DODO Qin et al. (2024) builds compressed representations of its past context dynamically. It is primarily intended as an efficient long-context method but it can be used for context compression. However, results when using DODO to compress multiple documents for RAG are weak (see Louis et al. (2025) Table 2.)

None of these methods can be used online and reach large compression rates. In fact, soft compression is merely succeeding with large compressors, and thus is really challenging with low-latency. OSCAR addresses this issue by using appropriate compressor backbone and training, as well as by computing query-dependent embeddings, which favor the task at hand.

## 3    METHOD

Figure 2 provides an overview of OSCAR. At inference, after retrieval, a *compressor* LLM maps each document-query pair to a few embedding tokens and a *generator* LLM generates an answer to the query based on the and a RAG prompt. Provided the compression rate is high and the compression operation efficient, there can be efficiency gains compared to the no-compression RAG pipeline. We now give details about every component of OSCAR as well as the training procedure.

**Compression**    The compression procedure is shown within Figure 2 (right). Contrary to Ge et al. (2023); Rau et al. (2024b); Louis et al. (2025), the document compression operation is conditioned on the query. In details, the query $q$, the $i$-th retrieved document $d_i$, a set of learnable memory tokens $[\texttt{MEM } j]_{j=1\ldots l}$ are fed forward to a *compressor* LLM $\mathcal{C}$. We collect the last layer hidden states corresponding to each of these tokens to form the query-dependent embedding representations $(c_i^1, \ldots, c_i^l) := \mathbf{c}_i = \mathcal{C}(q, d_i)$ of the document. $[\texttt{MEM } j]_j$ tokens play a similar role as the $[\texttt{CLS}]$ BERT token: it is a task-specific token prompting the storage within the corresponding hidden states.

**Generation**    The embedding representations $\mathbf{c}$ of each document, as well as the query $q$, are fed within a RAG prompt (given in Figure 11) to a *generator* LLM which generates the answer. Since each document is replaced by $l$ embeddings, generation is much faster compared to the original text.

**Compressor architecture**    All prior work on compression for RAG (Louis et al., 2025; Cheng et al., 2024; Rau et al., 2024b) use a compressor architecture identical to the generator LLM. In this setup, the hidden state representations of the compressor are easily adapted to the generator hidden space, making the whole pipeline easier to learn and deploy. But running the compression at inference

time would negate any subsequent generation time gains, making these methods inherently offline. OSCAR however is intended to operate in an online fashion with no possibility to pre-compute document compressions. Therefore, the compression needs to be fast. To do so, we propose two different architectures for the compressor backbone:

- **OSCAR-N-Layers**: we construct headless transformers using the first $N$ layers of the pretrained backbone (same architecture as the generator). As shown in §4.1, OSCAR-$N$-Layers models require no pre-training to align hidden representations with the generator LLM. Efficiency is controlled by the choice of $N$. We typically set $N$ to 1/4-1/3 the total number of layers.

- **OSCAR-llama**: we use a smaller LLM, primarily llama-1B[2], as our compressor. We apply two dense layers with ReLU non-linearity to the compressor last layer hidden space to align with the generator embedding space. Learning this mapping, which a crucial contribution of OSCAR, requires some pretraining (see Appendix Table 8) on top of the QA fine-tuning. Thus, following Rau et al. (2024b), we pretrain the compressor/generator LLM on auto-encoding and text-continuation tasks. Pretraining details are provided in Appendix H.

**Training objective**    The end-to-end OSCAR RAG pipeline should produce results as close as possible to its no-compression version. Therefore, we use a sequence-level distillation objective as in Louis et al. (2025): given a training set of questions and a collection of documents, we perform the retrieval stage and generate teacher labels from the standard no-compression RAG pipeline. These labels are then used as supervised-fine-tuning targets for the end-to-end OSCAR pipeline, as shown on Figure 2 (right). Overall, denoting $a_1, \ldots, a_r$ the answer generated by the teacher LLM from the documents and query, then the training objective on the compressor $\mathcal{C}$ and generator $\mathcal{G}$ is:

$$\mathcal{L}(\mathcal{C}, \mathcal{G}) = -\sum_{i=1}^{r} \log \mathcal{G}(a_i \mid q, \mathbf{c_1}, \ldots, \mathbf{c_k}, \mathbf{a_{<i}}), \text{ where } \mathbf{c_i} = (c_i^s)_{s=1,\ldots,l} = \mathcal{C}(q, d_i), \ i = 1, \ldots, k$$

(1)

where $k$ denotes the total number of documents used for generation. The loss is back-propagated both through the generator LLM and the compressor LLM at each step. Overall, OSCAR training does not require any ground truth labels. Initial experiments with the teacher choice and use of distillation objective gives identical conclusions to Louis et al. (2025): distillation is paramount and Mistral-7B labels offer good supervision. For simplicity, we use Mistral-7B as the teacher for all OSCAR models, whichever backbone they are based on. In practice, we save the retrieval results as well as teacher generations once on the training set so that OSCAR training is a simple supervised-fine-tuning between questions—augmented with document embeddings within the RAG prompt– and teacher answers. The subsequent OSCAR model training is fast: between 1 and 5 gpu-days for 1B-24B generator backbones.

**Simultaneous reranking**    Building on insights from Chirkova et al. (2025), query-dependent online context compression closely resembles document reranking. Rerankers, such as cross-encoders (Nogueira & Cho, 2019), refine the ranking from the initial retrieval step. Unlike retrieval models, which encode queries and documents independently, rerankers contextualize documents with respect to queries, yielding more informative representations. Since rerankers are already part of strong RAG pipelines Rau et al. (2024a), using a single forward-pass for both compression and reranking makes compression essentially free—so long as compression is no more expensive than typical rerankers.

We therefore add a reranking token [RR] to the compressor LLM prompt (Figure 2, right) and an additional dense layer which maps this token's hidden state to a predicted relevance score $r_i$. We train this added layer with a point-wise distillation objective from a reference reranker: we add $\lambda \sum_{i=1}^{k} (r_i - r_i')^2$ to equation 1, where $\lambda$ balances generation and reranking and $r_i'$ are scores from a reference reranker. While many training strategies exist (Hofstätter et al., 2021; Formal et al., 2022; Lin et al., 2021; Schlatt et al., 2024), simple point-wise distillation proved effective for OSCAR models.

---

[2]meta-llama/Llama-3.2-1B-Instruct

## 4 EXPERIMENTS

**Data** Our training dataset comprises questions from Louis et al. (2025) along with $500k$ queries extracted from MS MARCO (Nguyen et al., 2016), resulting in a total of $893k$ queries[3]. The document collection used for training is Wikipedia-KILT (Petroni et al., 2020), preprocessed into chunks of 128 tokens. Such chunking is typical in RAG pipelines (Rau et al., 2024a) and not a limitation as increasing the number of retrieved chunks still enables to extract long sequences of informative content. For each query, we retrieve the top-$k$ chunks using SPLADE-v3 (Formal et al., 2021; Lassance et al., 2024) and subsequently rerank them with a DeBERTa-v3 (He et al., 2021)-based reranker (a robust RAG setting as shown by Rau et al. (2024a)). We employ sentence-level distillation from Mistral-7B[4], as recommended by Louis et al. (2025).

**Training details** During training, the number $k$ of retrieved documents is set to 5. We empirically found that this value provides sufficient context for models to generalize to a larger number of documents at inference time while keeping training costs low. Each document is then compressed into $l$ embedding vectors, where $l$ is fixed for each OSCAR model. Specifically, OSCAR models with a compression rate of 16 use 8 memory embeddings per document – given 128-sized input documents. All generators LLMs are trained with LoRA Hu et al. (2021) adapters. For OSCAR-$N$-Layers models, we experiment with $N = 5, 8, 10$. OSCAR-llama relies on Llama-3.2-1B et al. (2024). All compressors are trained with full-fine tuning – which was consistently more effective than LoRA adapters. For joint training (§4.3), early experiments suggested that $\lambda = 0.05$ usually offers the best compromise (in terms of compression quality and reranking effectiveness) on the validation set – and we use this default value for all further corresponding experiments. Additional hyper-parameters are given in Appendix G.

**Baselines and Backbones** We compare OSCAR to Provence and Recomp models Chirkova et al. (2025); Xu et al. (2023) as they are the state-of-the-art hard compression models for RAG. We also run evaluations of PISCO models, a state-of-the-art offline soft compression model. Finally we provide a no-retrieval baseline as well as the performances of the no-compression RAG pipelines. Unlike most hard compression methods, OSCAR models are backbone-specific and need to be retrained for every different generation LLM. To show how stable OSCAR training is, we produce models for Mistral-7B-Instruct, Qwen2-7B-Instruct, Mistral-24B[5] and Llama-1B. We keep identical parameters/data/configurations for all backbones. Training times range between 1 to 4 GPU-days from 1B to 24B backbones.

For most of the experiments, we train OSCAR models without reranking ability. In §4.1, we provide evaluation metrics for OSCAR when compared to competitive approaches. In §4.2 we run ablations to identify the critical components of OSCAR. In §4.3, we show results of OSCAR models with reranking ability.

**Evaluation** After training, we evaluate all models on multiple datasets: Natural Questions Kwiatkowski et al. (2019), TriviaQA Joshi et al. (2017), HotpotQA Yang et al. (2018), ASQA Stelmakh et al. (2022), PopQA Mallen et al. (2022), and BIOASQ-12B Krithara et al. (2023). For each query, we retrieve documents from either KILT or PUBMED – a collection unseen during training. We measure three different evaluation metrics to ascertain the quality of OSCAR models:

(i) **accuracy**: for some question, accuracy is 1 if the (normalized) label is included in the (normalized) generated answer, where normalization is described in Appendix I.

(ii) **LLM evaluation**: we prompt an LLM to determine whether the predicted answer corresponds to the ground truth answer. This evaluation metric is robust to semantically-equivalent reformulation of the answer and better correlated to human judgements Kamalloo et al. (2023). Details in Appendix E

(iii) **pairwise-comparison using gpt-4o**: given generated answers from two models, we prompt gpt-4o to determine which answer is best, or if they are equivalent. Pairwise evaluation is a good complement to pointwise Zheng et al. (2023). Details in Appendix F.

---

[3]We will release the queries as well as the distillation labels upon publication

[4]huggingface/mistralai/Mistral-7B-Instruct-v0.2

[5]mistralai/Mistral-Small-24B-Instruct-2501

| Backbone | Compressor | Accuracy | | | | | | | Tera-Floating point operations | | |
|---|---|---|---|---|---|---|---|---|---|---|---|
| | | ASQA | HotpotQA | NQ | TriviaQA | POPQA | BIOASQ | Average | Inference | Compression | Total |
| **Mistral-7B** | No RAG | 0.51 | 0.34 | 0.46 | 0.79 | 0.29 | 0.40 | 0.47 | - | - | - |
| | No compression | 0.75 | 0.51 | 0.68 | 0.92 | 0.70 | 0.51 | **0.68** | 20.33 | 0. | 20.33 |
| | RECOMP | 0.73 | 0.49 | 0.67 | 0.92 | 0.67 | 0.53 | 0.67 | 7.29 | 0.84 | 8.13 **(2.5×)** |
| | Provence | 0.76 | 0.49 | 0.69 | 0.92 | 0.69 | 0.54 | **0.68** | 7.63 | 1.80 | 9.43 **(2.2×)** |
| | PISCO | 0.71 | 0.48 | 0.65 | 0.90 | 0.64 | 0.49 | 0.65 | 3.49 | offline | 3.49 **(5.8×)**[a] |
| | OSCAR-llama | 0.74 | 0.53 | 0.68 | 0.92 | 0.68 | 0.52 | **0.68** | 3.49 | 2.66 | 6.15 **(3.3×)** |
| | OSCAR-5-Layers | 0.73 | 0.50 | 0.66 | 0.91 | 0.66 | 0.50 | 0.66 | 3.49 | 3.04 | 6.53 **(3.1×)** |
| | OSCAR-8-Layers | 0.74 | 0.53 | 0.67 | 0.92 | 0.68 | 0.52 | **0.68** | 3.49 | 4.87 | 8.36 **(2.4×)** |
| **Llama-1B** | No compression | 0.61 | 0.35 | 0.54 | 0.82 | 0.59 | 0.40 | 0.55 | 2.85 | 0. | 2.85 |
| | OSCAR-5-Layers[b] | 0.64 | 0.43 | 0.59 | 0.86 | 0.59 | 0.46 | 0.60 | 0.50 | 0.88 | 1.38 **(2.1×)** |
| **Qwen-7B** | No compression | 0.70 | 0.51 | 0.64 | 0.90 | 0.64 | 0.53 | 0.65 | 18.94 | 0. | 18.94 |
| | OSCAR-8-Layers | 0.72 | 0.50 | 0.64 | 0.91 | 0.67 | 0.51 | 0.66 | 3.17 | 5.07 | 8.25 **(2.3×)** |
| | OSCAR-llama | 0.72 | 0.51 | 0.66 | 0.91 | 0.68 | 0.52 | **0.67** | 3.17 | 2.65 | 5.83 **(3.2×)** |
| **Mistral-24B** | No compression | 0.74 | 0.54 | 0.68 | 0.92 | 0.70 | 0.53 | 0.68 | 64.29 | 0. | 64.29 |
| | OSCAR–llama | 0.75 | 0.54 | 0.70 | 0.93 | 0.70 | 0.53 | **0.69** | 10.72 | 2.65 | 13.37 **(4.8×)** |

Table 1: **Accuracy and efficiency for OSCAR models and baselines based on various backbones.** OSCAR models are more effective and as accurate than their backbones with no compression. OSCAR models are also more efficient than the two hard compression baselines Provence and Recomp.

---

[a]PISCO is intended to be used offline and given for comparison.

[b]We do not train an OSCAR-llama with llama-3.2-1B backbone as it would not increase global efficiency.

Altogether, these metrics enable thorough evaluation and comparisons. OSCAR models have seen 5 retrieved documents per query at training time, but we evaluate them – and all other models – in a setting with 10 documents to verify generalization to larger contexts.

**Computational efficiency**    To evaluate computational efficiency, we sum the number of floating-point operations required for compression and for answer generation. For consistency, we perform this calculation on a standardized input query of 128 tokens, concatenated with 10 documents of 128 tokens each or their compressed embeddings. Measurements are obtained using `torch.profiler`. Further details, including computation times and peak memory usage, are provided in Appendix D.

## 4.1    MAIN RESULTS

Table 1 shows the accuracy results for all backbones, as well as efficiency measures. First, the no-RAG baseline has very low performance, indicating that these datasets are appropriate for RAG evaluations: the models cannot rely on memorization. Second, **OSCAR models are faster than hard compression baselines while preserving the accuracy of the no-compression models**. Using OSCAR models in-place of the underlying backbones enables a 2.2-4.8x inference speed-up. Among OSCAR variants, OSCAR-llama is generally the strongest and fastest, though it requires pretraining (see 4.2). Most interestingly, **OSCAR-llama model for Mistral-24B enables a 5× decrease in computational complexity while improving the overall results**. In fact, OSCAR efficiency improvements are proportional to the backbone size, and hence particularly advantageous for larger language models.

For OSCAR-$N$-Layers models, performance improves with more layers but at the cost of efficiency. Beyond 10 layers, accuracy plateaus while efficiency worsens (details in Appendix C).

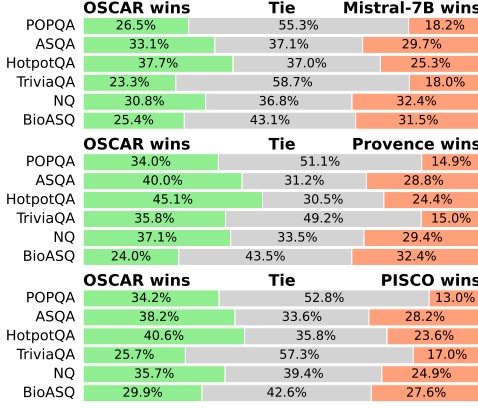

Figure 3: **GPT-4 pairwise comparisons.** OSCAR-llama, while faster, is on par with no compression baseline, Provence and PISCO.

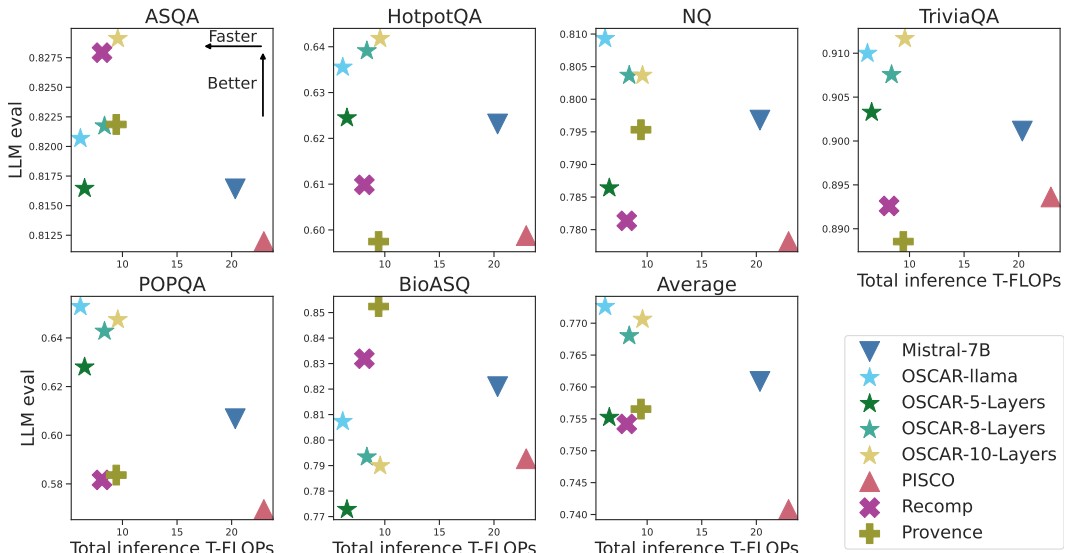

Figure 4: **LLM evaluation scores of each Mistral-7B-backboned models, in relation with the total number of floating point operations required at inference**. OSCAR models are faster and more effective on most datasets. OSCAR-llama in particular offers the best alternative. For PISCO, we include in the FLOPs the compression cost, as if it was used online.

Figure 4 shows LLM evaluation results for Mistral-7B models. These confirm the conclusions based on the accuracy metric. In fact, OSCAR models tend to be favorably appreciated by this LLM-evaluation. We hypothesize that since the retrieval is embedding-based rather than text-based for OSCAR, then reformulation of the answer into a semantically-equivalent answer is more likely to occur and to comparatively penalize the accuracy measure. Detailed results for all backbones are given on Table 3.

Finally, Figure 3 provides the results of pairwise comparisons of OSCAR-llama, Mistral-7B, PISCO and Provence. These confirm that OSCAR, while faster, is on par with its uncompressed baseline and slightly better on average than Provence. **Overall, OSCAR models offer an efficient alternative to regular RAG pipelines, with a x2-5 speed-up but little to no loss in accuracy.**

## 4.2 ABLATIONS

We run ablations to understand the effect of the components of OSCAR, with results shown on Table 2. All ablations use a Mistral-7B backbone.

**Query-dependence and compression rate.** First, Table 2 shows that accuracy losses with x128 compression are limited, with only 2% decrease on average. Second, we show that not using the query at compression leads to strong performance degradation, even more pronounced for large compression rate (-6%). Thus, OSCAR did succeed in using the query to optimize the compressed representation. Furthermore, in Appendix J, we look into the content of the compressed embeddings, to assess that they do indeed depend on the query. Figure 13 uses a needle-in-a-haystack test gkamradt (2024) to show that cosine similarity between compressed embeddings and text tokens is highest near the needle, indicating strong query dependence. Second, Figure 14 examines OSCAR embeddings via logit attributions nostalgebraist (2020), revealing that they align closely in vocabulary space with context relevant to the query.

**Freezing the generator** For OSCAR, we train jointly the generator and compressor models. We tried keeping the generator frozen, so as to allow to preserve fully the pretrained model in its state, but not succeed in obtaining satisfying performance, as shown on Table 2.

**Other compressor architectures** Results shown in §4.1 relied on Llama-1B as the compressor LLM. To obtain further efficiency gains, we tested using smaller compressors: modern-bert, modern-bert-large (Warner et al., 2024) and DeBERTa-v3 (He et al., 2021). Table 2 shows results after

| | Model | compr rate | ASQA | HotpotQA | NQ | TriviaQA | POPQA | Avg (Δ) |
|---|---|---|---|---|---|---|---|---|
| **Compression rate 16** | OSCAR-llama | x16 | 0.82 | 0.64 | 0.81 | 0.91 | 0.65 | 0.77 |
| | query-independent | x16 | 0.81 | 0.60 | 0.78 | 0.89 | 0.57 | 0.73 (-0.04) |
| | no compressor pretraining | x16 | 0.78 | 0.56 | 0.75 | 0.89 | 0.51 | 0.70 (-0.07) |
| | frozen generator | x16 | 0.78 | 0.54 | 0.76 | 0.88 | 0.60 | 0.71 (-0.06) |
| **Compression rate 128** | OSCAR-llama | x128 | 0.81 | 0.61 | 0.79 | 0.90 | 0.63 | 0.75 (-0.02) |
| | query-independent | x128 | 0.81 | 0.57 | 0.75 | 0.89 | 0.51 | 0.71 (-0.06) |
| **Other compressor architectures** | DeBERTa-v3 | x16 | 0.80 | 0.61 | 0.77 | 0.90 | 0.57 | 0.73 (-0.04) |
| | Modern-bert-base | x16 | 0.80 | 0.62 | 0.77 | 0.90 | 0.60 | 0.74 (-0.03) |
| | Modern-bert-large | x16 | 0.83 | 0.63 | 0.80 | 0.91 | 0.64 | 0.76 (-0.01) |
| **BM25 retrieval pipeline** | No compression | - | 0.57 | 0.56 | 0.57 | 0.81 | 0.37 | 0.58 |
| | OSCAR-llama | x16 | 0.57 | 0.52 | 0.56 | 0.80 | 0.37 | 0.56 (-0.02) |

Table 2: **Ablation study** on compression rate, pretraining, compressor architectures and retrieval pipeline. The last column reports averages across the five QA tasks, and the difference compared to OSCAR-llama (x16). We report point-wise LLM evaluation.

pretraining and fine-tuning with different compressors. Llama-1B performs the best. Modern-bert-large may offer an interesting alternative for low-latency applications.

**Robustness to retrieval changes.** In all training and test experiments so far, all documents were retrieved using SPLADE-v3 and reranked with a DeBERTa-v3-based reranker – a robust RAG setup Rau et al. (2024a). Yet it still prompts the question of how OSCAR models perform when retrieval quality declines. In particular, the behavior of hard compression methods is clearly identifiable on noisy documents – and shown to be correctly handled by Provence Chirkova et al. (2025) or Recomp Xu et al. (2023). It is more of an open question for soft compression models like OSCAR. To investigate this, we run evaluation experiments using BM25 Robertson et al. (1996) only (no reranking) and report results on Table 2. Essentially, the performance drops of OSCAR models with respect to Mistral-7B are similar – indicating that OSCAR models are able to handle noisy documents. Detailed results for all datasets are found in Appendix B.

**Long context abilities of OSCAR models.** Since OSCAR models are trained with 5 retrieved documents, we investigate whether they remain able to extract and use information from a larger number of documents. Figure 5 shows the results when increasing the number of retrieved documents to up to 50 (which makes uncompressed contexts around $7k$ tokens) on ASQA. Note that as the number of documents increase, because of the quadratic cost of the attention, the larger compression rate of OSCAR models make them comparatively faster. With 50 documents, we measure 5× less FLOPs for OSCAR than Mistral-7B. Further analysis on the ability of OSCAR to compress longer documents is in Appendix K.

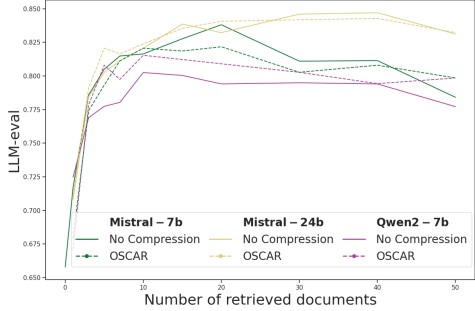

Figure 5: LLM evaluations with increasing number of retrieved documents: OSCAR models are as robust as their no-compression baselines.

### 4.3 ADDING RERANKING CAPABILITY

Having demonstrated that OSCAR models function effectively as standalone compressors, we also train OSCAR models capable of both document compression and reranking. In a RAG pipeline incorporating reranking, the computational cost of compression becomes virtually negligible, as a single forward pass produces both compressed representations and reranking scores.

The results in Table 4 in the appendix show the performance of such jointly trained models under two evaluation settings: standalone, which corresponds to the previous setting (DeBERTa-v3 reranker), and e2e which corresponds to compressing documents reranked by the OSCAR model itself. Essentially, we observe no drop in performance between standalone and e2e settings, indicating that OSCAR effectively learns to rerank documents. This finding is further supported by OSCAR's

performance on the BEIR benchmark Thakur et al. (2021) where its reranking capabilities are nearly on par with the strong teacher model. Detailed BEIR results for individual datasets are provided in Appendix (Table 5). To match the teacher's performance on BEIR, OSCAR requires an increased model depth to 16 layers. However, this model is less efficient, and its actual e2e performance (evaluated via LLM-based metrics or accuracy) remains unchanged.

## 5  CONCLUSION

In this paper, we introduce OSCAR, the first online soft compression methods for RAG. The key challenge is designing an efficient compression technique for an online setting, which we address with two variants: one using a small compressor model and another leveraging the generator's early layers. We compare OSCAR against hard compression methods (RECOMP, Provence) and soft ones (PISCO), showing that query-dependent compression is more effective than query-independent approaches. OSCAR also outperforms or matches hard pruning methods while being more efficient, proving the potential of soft compression. Additionally, we extend OSCAR with reranking, thereby reducing compression costs by factorization in the RAG pipeline. Our ablations analyze different backbones, weak retriever performance, behavior with large number of retrieved documents and further validate the design and performance of OSCAR models.

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

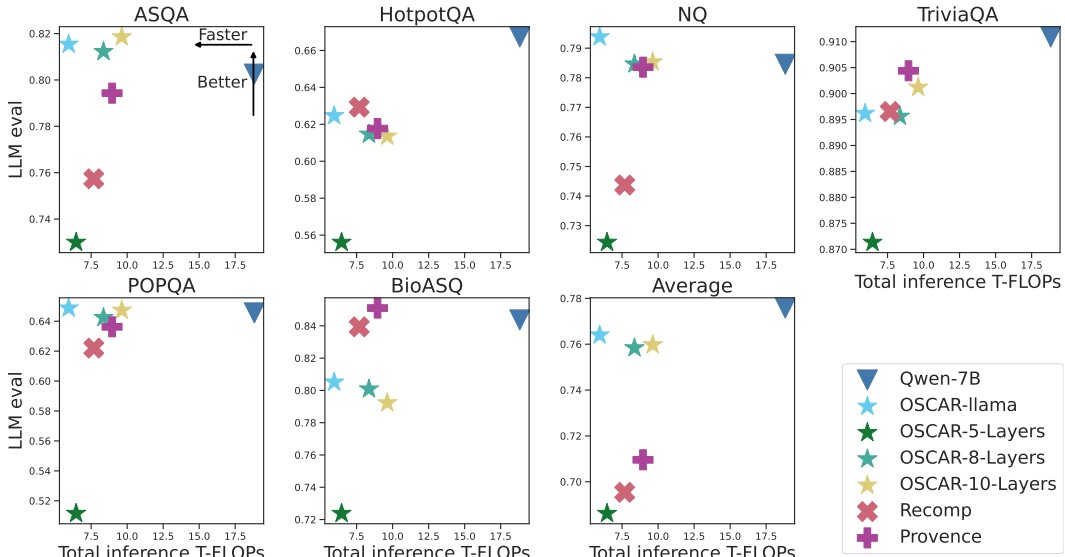

Figure 6: **LLM evaluation of Qwen2-7B-backboned models**, in relation with the total number of floating point operations required at inference. OSCAR-llama model is the fastest and best compression model.

## A    ADDITIONAL RESULTS

### A.1    OSCAR WITH QWEN2-7B

We showed in Figure 4 the efficiency/performance plots for Mistral-7B backbone, including comparison with Provence, Recomp and the uncompressed backbone. We provide in Figure 6 the same results but for Qwen2-7B. OSCAR-llama models remains the best compression model, both in terms of efficiency and LLM evaluation score. In particular, OSCAR-llama score is on average 4 points above Provence and 6 points above RECOMP.

### A.2    DETAILED LLM EVALUATION RESULTS

In Section 4.1, we provided LLM evaluation results for Mistral-7B models. On Table 3 shows all LLM-evaluation results. In Section 4.1, we provided pareto plot efficiency/LLM evaluation for Mistral-7B Backbone. We provide on Figure 7 the corresponding efficiency/accuracy pareto plot. Conclusions are mostly identical to the main results

### A.3    FULL RESULTS ON THE BEIR DATASET

We report in Table 5 the detailed BEIR results on individual datasets.

## B    DETAILED EFFECTS OF BM25 RETRIEVAL

In section 4.2, we provided averaged effect across datasets of the change of retrieval/reranking pipeline. We provide in Figure 8 results for individual datasets. These results show that performance is preserved across all datasets, although it is likely that retrieval for Bioasq is noisier.

## C    INFLUENCE OF NUMBER OF COMPRESSOR LAYERS

In Section 3, we proposed constructing a transformer by utilizing the initial layers of the backbone to develop an efficient compressor that operates without requiring pretraining. Since the inference cost scales with the number of retained layers, it is important to examine the impact of reducing the

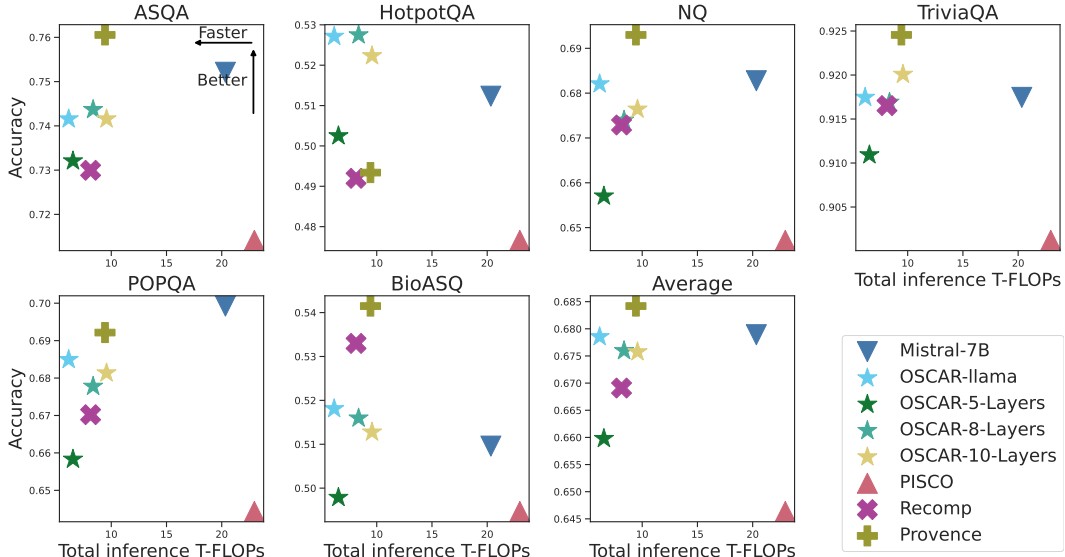

Figure 7: **Accuracy scores of each Mistral-7B-backboned models**, in relation with the total number of floating point operations required at inference. OSCAR models are faster and better on most datasets.

| Backbone | Compressor | ASQA | HotpotQA | NQ | TriviaQA | POPQA | BIOASQ | Average |
|---|---|---|---|---|---|---|---|---|
| **Mistral-7B** | No compression | 0.82 | 0.62 | 0.80 | 0.90 | 0.61 | 0.82 | 0.76 |
| | RECOMP | 0.83 | 0.61 | 0.78 | 0.89 | 0.58 | 0.83 | 0.75 |
| | Provence | 0.82 | 0.60 | 0.80 | 0.89 | 0.58 | 0.85 | 0.76 |
| | PISCO | 0.81 | 0.60 | 0.78 | 0.89 | 0.57 | 0.79 | 0.74 |
| | OSCAR-llama | 0.82 | 0.64 | 0.81 | 0.91 | 0.65 | 0.81 | 0.77 |
| | OSCAR-5-Layers | 0.82 | 0.62 | 0.79 | 0.90 | 0.63 | 0.77 | 0.76 |
| | OSCAR-8-Layers | 0.82 | 0.64 | 0.80 | 0.91 | 0.64 | 0.79 | 0.77 |
| **Llama-1B** | No compression | 0.69 | 0.48 | 0.66 | 0.81 | 0.52 | 0.76 | 0.65 |
| | OSCAR-5-Layers[a] | 0.71 | 0.53 | 0.70 | 0.85 | 0.55 | 0.72 | 0.68 |
| **Qwen-7B** | No compression | 0.80 | 0.67 | 0.78 | 0.91 | 0.65 | 0.84 | 0.78 |
| | OSCAR-8-Layers | 0.81 | 0.61 | 0.78 | 0.90 | 0.64 | 0.80 | 0.76 |
| | OSCAR-llama | 0.82 | 0.62 | 0.79 | 0.90 | 0.65 | 0.81 | 0.76 |
| **Mistral-24B** | No compression | 0.82 | 0.71 | 0.80 | 0.92 | 0.70 | 0.85 | 0.80 |
| | OSCAR–llama | 0.82 | 0.65 | 0.82 | 0.92 | 0.67 | 0.84 | 0.79 |

Table 3: **LLM evaluation and efficiency for OSCAR models and baselines based on various backbones.** OSCAR models are more effective and faster than their backbones with no compression. OSCAR models are also more efficient than the two hard compression baselines Provence and Recomp.

---

[a]We do not train an OSCAR-llama with llama-32-1B backbone as it would not increase global efficiency.

number of layers used for compression. This analysis is presented in Figure 9, where the performance appears to plateau around 4-5 layers for Mistral-7B. Notably, increasing the number of layers beyond 10 does not seem to justify the additional computational cost.

| Model | Setting | LLM evaluation score | | | | | | | BEIR |
| | | ASQA | HotpotQA | NQ | TriviaQA | POPQA | BIOASQ | Average | |
|---|---|---|---|---|---|---|---|---|---|
| **OSCAR-llama** | standalone | 0.83 | 0.64 | 0.80 | 0.91 | 0.66 | 0.80 | 0.77 | 52.8 |
| | e2e | 0.81 | 0.63 | 0.79 | 0.91 | 0.66 | 0.80 | 0.77 | |
| **OSCAR-8-Layers** | standalone | 0.82 | 0.64 | 0.81 | 0.91 | 0.64 | 0.79 | 0.77 | 52.5 |
| | e2e | 0.81 | 0.63 | 0.79 | 0.90 | 0.64 | 0.78 | 0.76 | |
| **OSCAR-10-Layers** | standalone | 0.82 | 0.64 | 0.81 | 0.91 | 0.64 | 0.80 | 0.77 | 54.3 |
| | e2e | 0.81 | 0.65 | 0.82 | 0.91 | 0.66 | 0.78 | 0.77 | |

Table 4: **LLM evaluation and reranking performance on the BEIR benchmark (mean nDCG@10 on the 13 BEIR datasets)**. We report results for three efficient OSCAR models on two RAG settings (with a Mistral-7B decoder). The reranking performance of the teacher (based on DeBERTa-v3) is 55.4. Note that the performance on the standalone setting might slightly differ from previous Tables as these models are trained with a different loss (joint training).

| Corpus | DeBERTa-v3 | OSCAR-llama | OSCAR-8-Layers | OSCAR-10-Layers | OSCAR-16-Layers |
|---|---|---|---|---|---|
| TREC-COVID | 88.3 | 83.1 | 81.4 | 84.4 | 86.1 |
| NFCorpus | 37.5 | 34.2 | 34.5 | 36.5 | 36.9 |
| NQ | 66.7 | 63.3 | 61.3 | 64.1 | 67.2 |
| HotpotQA | 74.5 | 72.9 | 72.2 | 73.5 | 74.3 |
| FiQA-2018 | 47.8 | 42.7 | 40.8 | 44.3 | 47.5 |
| ArguAna | 29.8 | 29.5 | 32.5 | 32.4 | 34.0 |
| Touché-2020 | 33.5 | 29.3 | 31.6 | 31.9 | 31.3 |
| Quora | 84.8 | 86.0 | 86.0 | 87.5 | 87.9 |
| DBPedia | 48.9 | 47.5 | 46.5 | 48.2 | 49.2 |
| SCIDOCS | 19.2 | 17.2 | 17.6 | 18.6 | 19.3 |
| FEVER | 86.6 | 83.6 | 83.1 | 84.1 | 83.9 |
| Climate-FEVER | 27.4 | 25.9 | 24.2 | 25.3 | 26.3 |
| SciFact | 75.8 | 71.2 | 71.2 | 75.2 | 75.5 |
| average | 55.4 | 52.8 | 52.5 | 54.3 | 55.3 |

Table 5: **nDCG@10 on the 13 open BEIR datasets**. DeBERTa-v3 is the reranker teacher used to train OSCAR models.

# D  MORE ABOUT EFFICIENCY

## D.1  SETUP TO MEASURE EFFICIENCY

In Section 4.1, we measured efficiency of models based on the total number of floating-point operations as it is the primary indicator of the computational complexity. To generate these measures, we generate fake inputs of standardized size (a query/prompt of 128 tokens associated with 10 128-token documents) and do compression and the generation of a 32 token answer[6] from an input of size computed from the compression rate of each method (e.g., for OSCAR with compression rates 16, the input to the generator is of size $128 + 10\frac{128}{16}$). To compute FLOP we set the batch size to 1 and use `torch.profiler`. We provide additional measures regarding inference time and peak GPU memory in each case. We set the batch size at 256 (32 for the larger Mistral-24B) to compute the inference time (simulating a busy service) and the peak GPU memory. In all cases we use hugging face implementation of the models. For memory usage and inference time, we average the results over 10 runs.

Results are shown in Table 6. Gains observed in terms of floating-point operations mostly translate to computational time (as can be expected for sufficiently large batch sizes). OSCAR models enable to save about 50-75% of memory across the various backbones. In practice, this larger batch sizes to be used and hence further latency improvements.

---

[6]The analysis for generated answers of 128 or 256 tokens leads to similar conclusions

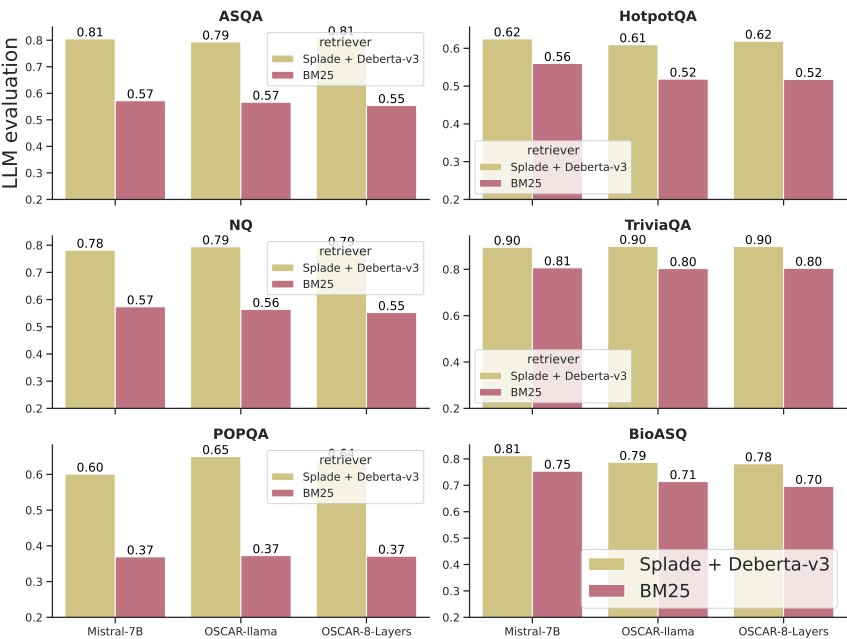

Figure 8: Effect of retrieval on OSCAR models, per dataset, compared to their uncompressed backbone.

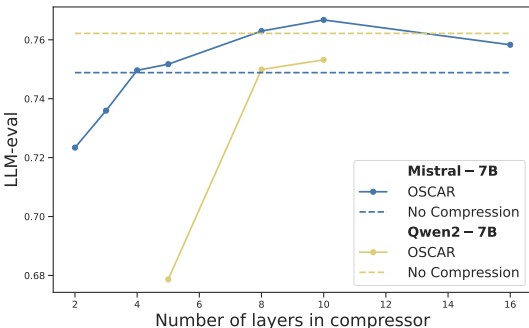

Figure 9: Average accuracy on general domain datasets for OSCAR models where the compressor has a variable number of layers. Performances increase with the number of layers but plateau above 8-10 layers for both Qwen2-7B and Mistral-7B backbones.

# E    LLM EVALUATION

Our primary evaluation metric follows the LLM-based assessment proposed in Rau et al. (2024a). This approach utilizes the SOLAR-107B model[7] prompted to determine the correctness of a predicted answer by comparing it against both the given question and a reference answer. This metric can be viewed as an enhanced version of traditional accuracy, as it remains more robust to surface-level variations that do not alter the underlying semantic content. The prompt used is given in Figure 10.

# F    PROMPTS

The prompt we use for generation is given on Figure 11. The prompt for GPT pairwise comparison is given on Figure 12

---

[7]huggingface/upstage/SOLAR-10.7B-Instruct-v1.0

Figure 10: LLM Evaluation Prompt

**system**: "You are an evaluation tool. Answer with one of 1: Correct, 0.5: Partially correct, 0: wrong.
**user**: "Here is a question, a golden answer, and an AI-generated answer. Can you judge whether the AI-generated answer is correct according to the question and golden answer? Simply answer with one of 1: correct, 0.5: partially correct, 0: wrong. Question: {question}. Golden answer: {answer}. Generated answer: {prediction}."

Figure 11: Main Prompt

**system**: You are a helpful assistant. Your task is to extract relevant information from provided documents and to answer questions as briefly as possible.
**user**: Background:
{doc$_1$}SEP{doc$_2$}...SEP{doc$_k$}
Question: {question}

Figure 12: Gpt-4o Pairwise Comparison Prompt

**system**: "You are a helpful assistant that ranks models by the quality of their answers. Please act as an impartial judge. Do not allow the length of the responses to influence your evaluation. Be as objective as possible."
**user**: "Here is a question, a ground truth answer, an AI-generated answer 1, and an AI-generated answer 2. Which answer is the most correct one? Simply answer 1 if the first is better, 2 if the second is better, and 3 if it's a tie.
Question: {question}.
Ground truth answer: {ref answer}.
Answer 1: {answer$_1$}.
Answer 2: {answer$_2$}."

| Backbone | Compressor | | Inference time (ms)[†] | | | Peak memory (Gb)[‡] |
| | Architecture | Parameters | Inference | Compression | Total | |
|---|---|---|---|---|---|---|
| **Mistral 7B** | No compression | - | 141.6 | 0. | 141.6 | 24.3 |
| | OSCAR-5L | 1.2B | 33.0 | 18.0 | 51.0 (**2.3×**) | 16.2 |
| | OSCAR-8L | 1.91B | 33.0 | 28.8 | 61.8 (**2.2×**) | 16.2 |
| | OSCAR-llama | 1.1B | 33.0 | 17.1 | 50.1 (**2.8×**) | 16.2 |
| **Llama 3.2 1B** | No compression | - | 30.2 | 0. | 30.2 | 8.6 |
| | OSCAR-5L | | 8.3 | 5 | 13.3 (**2.3×**) | 4.3 |
| **Qwen-2-7B** | No compression | - | 109 | 0. | 109 | 30.2 |
| | OSCAR-5L | 1.7B | 25.6 | 15.2 | 40.8 (**2.7×**) | 23.3 |
| | OSCAR-llama | 1.1B | 25.6 | 17.1 | 42.7 (**2.6×**) | 23.3 |
| **Mistral-24B** | No compression | - | 383.2 | 0. | 383.2 | 69.2 |
| | OSCAR–llama | 1.1B | 67.9 | 17.1 | 85.0 (**4.5×**) | 51.9 |

Table 6: **Inference time and memory for each model**. Computed with 128-token queries and 10 128-token retrieved documents. [†] computed with batch size 256 (32 for Mistral-24B) but brought down to individual query cost [‡]for a batch of size 32.

| Hyperparameter | Value |
|---|---|
| Batch Size | 128 |
| LR generator | $1 \times 10^{-4}$ |
| LR llama compressor | $1 \times 10^{-4}$ |
| LR N-layers compressor | $5 \times 10^{-5}$ [a] |
| LR scheduler | linear |
| Optimizer | AdamW |
| Epochs | 1 |
| Max Tokens Teacher Generation | 128 |
| LoRA Layers ($r$) | all-linear |
| LoRA Rank ($r$) | 16 |
| LoRA Dropout | 0.1 |
| LoRA Alpha | 32 |
| Llama compressor hidden dim | 8096 |
| Weight Decay | 0.1 |
| Warmup Ratio | 0.05 |
| Max Gradient Norm | 1.0 |
| Documents max tokens | 128 |

Table 7: Fine-tuning Hyper-parameters.

---

[a]Initial results with identical learning rates between the LoRA-trained decoder and fully fine-tuned N-layers compressor gave poor results: learning rates need to be differentiated between compressor and decoder in this case.

## G  OSCAR TRAINING HYPERPARAMETERS

We provide in this section details enabling the replication of OSCAR training results. Note that all OSCAR models for all backbones (from llama-1B all the way to mistral-24B) were trained using this configuration. Our training code relies on HuggingFace trainer and an adaptation of the public Bergen library Rau et al. (2024a).

Note that OSCAR-N-layer models are directly trained by fine-tuning on the distillation data described in Section 4: they do not need pretraining. This is a similar effect as in Louis et al. (2025). On the contrary, OSCAR-llama models need a pretraining described in Appendix H.

**Hyper-parameter search to build OSCAR models**   We took hyperparameters from Louis et al. (2025) and only conducted a small grid search over 8 values to tune the learning rate required on the compressor, as we noticed performances were underwhelming with identical learning rates on

| Model | ASQA | HotpotQA | NQ | TriviaQA | POPQA |
|---|---|---|---|---|---|
| OSCAR-llama | 0.82 | 0.64 | 0.81 | 0.91 | 0.65 |
| + without pretraining | 0.78 | 0.56 | 0.75 | 0.89 | 0.51 |

Table 8: **Ablation on the pretraining for OSCAR-llama model.**

compressor and generator. The total computation time to train an OSCAR model around Mistral-7B is around 50 hours on a single high-end GPU.

## H  OSCAR-LLAMA PRETRAINING

OSCAR models using llama-1B as compressor models without any pretraining failed to reach satisfying performances (see Table 8). We attribute this effect to the need of building a map between the compressor hidden space and the decoder hidden space. To achieve this, we use the same pretraining as proposed in Rau et al. (2024b), with identical hyperparameters and a pretraining dataset consisting of chunks preprocessed from fineweb[8]. Note that experiments show that as long as some form of extended pretraining is done which requires the decoder to use embeddings produced by the compressor, the ensuing OSCAR-llama models are strong. Therefore, the exact recipe of the pretraining is not crucial for replicating our work.

## I  STRING NORMALIZATION FOR METRIC COMPUTATION

To measure accuracy, F1 score or recall between a ground truth label and a prediction, we check that the normalized label is included in the normalized prediction. When multiple labels are possible, we take maximum values across the available labels. Normalization consists in:

- Converting the string to lowercase
- Removing punctuation
- Removing articles: "a", "an", "the"
- Standardizing word splits by replacing multiple spaces and line returns with a single space

## J  RELATION BETWEEN EMBEDDINGS AND QUERY

While OSCAR offers greater computational efficiency and accuracy, it lacks the interpretability of hard compression methods. In this section, we offer a glimpse into the content of the compressed embeddings, to assess that they do indeed depend on the query. First, Figure 13 uses a needle-in-a-haystack test gkamradt (2024) to show that cosine similarity between compressed embeddings and text tokens is highest near the needle, indicating strong query dependence. Second, Figure 14 examines OSCAR embeddings via logit attributions nostalgebraist (2020), revealing that they align closely in vocabulary space with context relevant to the query.

## K  COMPRESSING LONGER DOCUMENTS

So far we systematically compressed 128-token documents. This offers a valid RAG pipeline with excellent performances. However, to further understand the robustness of OSCAR, we run additional experiments where we expose the model to 256-token documents:

- **DocMerge-5×2**: the top 10 retrieved documents are concatenated pairwise, producing 5 documents whose lengths are doubled. This setting isolates the model's ability to exploit longer but noise-free context.

---

[8]huggingface./datasets/HuggingFaceFW/fineweb

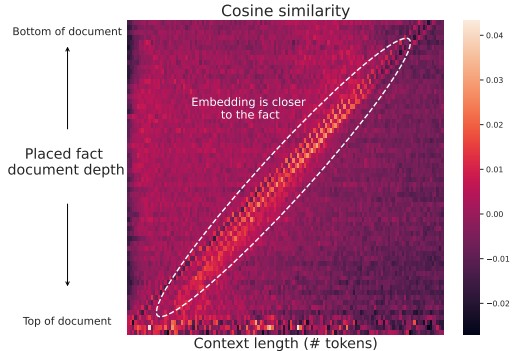

Figure 13: Cosine similarity between document embeddings and document individual tokens, on a needle-in-a-haystack test. The document embeddings are more similar to the area around the needle, indicating that the compression focuses on query-related elements.

Figure 14: Logits attributions on OSCAR embeddings. Attributed tokens predominantly correspond to an area of the context relevant to the query.

- **DocMerge-Noisy-10×2**: each of the top 10 retrieved documents is concatenated with a randomly selected irrelevant document, yielding 10 new documents twice as long. This evaluates robustness to increased context length combined with noise.

| Model | ASQA | HotpotQA | NQ | TriviaQA | POPQA | BioASQ12B | Avg. |
|---|---|---|---|---|---|---|---|
| Mistral-7B | 0.752 | 0.512 | 0.683 | 0.917 | 0.699 | 0.510 | 0.679 |
| Mistral-7B (DocMerge-5×2) | 0.732 | 0.484 | 0.650 | 0.911 | 0.649 | 0.495 | 0.654 |
| Mistral-7B (DocMerge-Noisy-10×2) | 0.674 | 0.513 | 0.682 | 0.918 | 0.696 | 0.502 | 0.664 |
| OSCAR-LLaMA | 0.742 | 0.527 | 0.682 | 0.917 | 0.685 | 0.518 | 0.679 |
| OSCAR-LLaMA (DocMerge-5×2) | 0.703 | 0.494 | 0.648 | 0.904 | 0.615 | 0.495 | 0.643 |
| OSCAR-LLaMA (DocMerge-Noisy-10×2) | 0.715 | 0.503 | 0.651 | 0.909 | 0.648 | 0.510 | 0.656 |

Table 9: Performance of OSCAR-LLaMA and its Mistral-7B backbone under long-context robustness settings.

**Discussion.** On the **DocMerge-5×2** condition, OSCAR—despite not being trained on 256-token documents—exhibits only a modest performance drop (approximately $-1\%$ on average) and still achieves competitive accuracy across benchmarks. In the **DocMerge-Noisy-10×2** setting, where longer context is coupled with injected noise, OSCAR performs on par with its Mistral-7B backbone (within $0.7\%$), highlighting its strong ability to process and utilize extended evidence even under noisier long-context regimes.

