# OpenReview forum: "OSCAR: Online Soft Compression for RAG"
_ICLR.cc/2026/Conference — ICLR 2026 Poster_

### Official Review · Reviewer_kf7i · 2025-10-31

**Soundness:** 4
**Presentation:** 3
**Contribution:** 3
**Rating:** 8
**Confidence:** 3

**Summary:**

This paper proposes OSCAR, an online, query-dependent soft compression mechanism for RAG. A compressor LLM maps each (query, document) pair to a small number of learned “memory” embeddings, which a generator LLM then consumes to answer the query. Two compressor backbones are explored: (i) OSCAR-N-Layers, using the first N layers of the generator; and (ii) OSCAR-llama, a small LLM with a learned hidden-space mapping. Across Mistral-7B, Qwen-7B, Llama-1B and Mistral-24B backbones, OSCAR yields 2–5× end-to-end speedups with minimal accuracy loss.

**Strengths:**

1. Propose a novel online soft compression method that compresses the input in a query-dependent fashion.

2. The training pipeline of OSCAR operates in a self-supervised way that does not rely on ground truth labels and generates them from no-compression RAG pipeline, providing flexibility over document choices.

3. Comprehensive evaluation with different choices of datasets, backbones and evaluation metrics. The results show a better accuracy-efficiency tradeoff for OSCAR over existing methods.

4. Detailed ablations on compression rate, query dependence, context length and compressor architecture.

**Weaknesses:**

1. OSCAR must be retrained per generator LLM to align hidden spaces; while the paper notes a few GPU-days for training, this could be operationally costly for trying out different backbone LLMs for different tasks.

2. Under some dataset setups OSCAR has worse accuracy compared to existing compression methods or the original backbone models. The paper does not provide a good investigation into the reasons causing this.

3. The baselines chosen by the paper are mainly hard compression and offline soft compression methods. There is a lack of comparison with existing online soft compression methods.

**Questions:**

1. How does OSCAR compare to existing online soft compression approaches such as FiD-light mentioned in the prior work?

2. In Figure 4 for BioASQ, why is the accuracy of OSCAR consistently worse than the baselines with the same compute T-FLOPs?

3. In Figure 3 for the comparison of OSCAR with Mistral-7B, OSCAR also does not show clear wins as compared to Provence and PISCO. What are the possible reasons behind this?

4. For the N value taken in OSCAR-N-Layers, what is the sensitivity of it when the backbone model changes?

---

> ### Author Response · Authors · 2025-11-19
>
> Thank you for your nice review. We are glad that you value our contributions. We address your comments below.
>
> > OSCAR must be retrained per generator LLM to align hidden spaces; while the paper notes a few GPU-days for training, this could be operationally costly for trying out different backbone LLMs for different tasks.
>
> Our study also shows that OSCAR performance follows the performance of the underlying backbone. Therefore, such an analysis can be conducted without training OSCAR for every backbone: only for the appropriate one.
>
> > The baselines chosen by the paper are mainly hard compression and offline soft compression methods. There is a lack of comparison with existing online soft compression methods.
>
> Directly comparable methods are RECOMP, Provence and PISCO. We did try a KV compression method which is adaptable to RAG via a smart chunk size choice. Latency improvements for this kind of method are obtained only at long context lengths (>4k) (since there is preprocessing cost to the KV cache pruning/compression) and we observed performance degradation on our datasets
> | Model                            | ASQA  |
> |----------------------------------|-------|
> | Mistral-7B-Chunk-Press, compression 4x      | 0.681 |
> | Mistral-7B-Chunk-Press, compression 2x       | 0.691 |
> | Mistral-7B-Chunk-Press, compression 1.3x     | 0.659 |
> | Mistral-7B-Instruct-v0.2, no compression        | 0.739 |
> Besides, such methods may be used ON TOP of OSCAR at generation and thus are kind of orthogonal in the way they improve efficiency. We will add this discussion in the paper.
> We also discussed DODO https://arxiv.org/abs/2310.02409 (and will add it in a revised version of the paper) in our response to 4qhT.
>
> > How does OSCAR compare to existing online soft compression approaches such as FiD-light mentioned in the prior work?
>
> FiD-light has a very low compression rate (it produces 50 embeddings per token). As discussed above, DODO reaches weak performance and is not designed specifically for RAG. We ran experiments with some KV cache methods (see right above) but could not obtain satisfying results in the RAG regime we are working with.
>
> > In Figure 4 for BioASQ, why is the accuracy of OSCAR consistently worse than the baselines with the same compute T-FLOPs?
>
> This is something we noticed as well. Since then, we ran additional out-of-domain evaluations to eliminate a potential overfitting on the general domain. We used RobustQA test suite which covers multiple domains and obtained for Mistral-24B:
>
> | model_name                                   |   RobustQA_Lifestyle |   RobustQA_Writing |   RobustQA_Science |   RobustQA_Recreation |   RobustQA_Technology |   average |
> |:--------------------------------------------|---------------------:|-------------------:|-------------------:|----------------------:|----------------------:|---------:|
> | Mistral-Small-24B-Instruct-2501             |                0.866 |              0.869 |              0.833 |                 0.779 |                 0.894 |    0.848 |
> | Oscar-llama 24B                             |                0.858 |              0.878 |              0.802 |                 0.751 |                 0.885 |    0.835 |
>
> These show that Oscar-llama is competititve with its backbone on other domains as well.
>
>
> > For the N value taken in OSCAR-N-Layers, what is the sensitivity of it when the backbone model changes?
>
> This is shown on Figure 9 in the appendix. Oscar built on Qwen2-7B with 8 layers is on par with OSCAR-llama, at 74% of LLMEval compared to 76% for the base model. A further study on mistral-24B would be interesting to confirm at larger scale.

---

### Official Review · Reviewer_j7H9 · 2025-10-31

**Soundness:** 3
**Presentation:** 3
**Contribution:** 3
**Rating:** 6
**Confidence:** 4

**Summary:**

The paper proposes OSCAR, a query-dependent, online soft compression framework for RAG. Unlike prior hard or soft compression approaches, OSCAR compresses each (query, document) pair into a small set of embedding tokens on the fly and conditions generation on these embeddings. Across open-domain QA benchmarks, OSCAR targets 2–5× end-to-end inference FLOPs reduction with near-parity accuracy to uncompressed RAG, with stronger gains on larger backbones.

**Strengths:**

- The method is elegantly designed, conceptually bridging the gap between traditional hard compression and offline soft compression in terms of both performance and usability.
- Strong empirical results, achieving near-parity accuracy with up to 2–5× efficiency gains.
- The integration with reranking allows shared computation, improving system efficiency without additional overhead.

**Weaknesses:**

1. My main concern is that fine-tuning the generator may impair generalization. Because the compressor is query-dependent, the resulting features can encode domain-specific priors; joint training encourages the generator to exploit these priors, increasing the chance of specialization and consequent degradation on cross-domain retrieval and non-RAG tasks. This potential loss of generality could limit the method’s practicality in real-world deployments.

2. The experiments are mainly conducted on Mistral-7B and Qwen2-7B, while larger models such as Mistral-24B are only compared against an uncompressed version. Moreover, the evaluation focuses on accuracy and FLOPs, omitting critical system-level metrics such as end-to-end latency and throughput. Although Table 6 reports related values, it lacks direct comparisons to baseline methods.

3. The paper claims that query-dependent embeddings preserve task-relevant semantics but provides limited theoretical analysis or formal justification to support this claim.

**Questions:**

- How does the LoRA-fine-tuned generator perform on standard non-RAG benchmarks ? Does fine-tuning lead to a degradation in general-purpose reasoning or language understanding?

- The current cross-domain evidence is limited, focusing mainly on biomedical QA and retrieval-level BEIR, without end-to-end evaluation in other domains or non-QA RAG tasks. How does the proposed approach behave on cross-domain retrieval or non-QA RAG tasks?

- Could the authors report results when the generator is kept frozen during training to isolate the effect of compression?

- Providing a more comprehensive evaluation would offer a fuller assessment of the method’s contribution and generality.

---

> ### Author Response · Authors · 2025-11-19
>
> Thank you for your feedback. We do share your opinion that the method is nicely designed and now well tested. We address your comments below.
>
> > The current cross-domain evidence is limited, focusing mainly on biomedical QA and retrieval-level BEIR, without end-to-end evaluation in other domains or non-QA RAG tasks. How does the proposed approach behave on cross-domain retrieval or non-QA RAG tasks
>
> - Regarding cross-doman retrieval performances, we made additional evaluations on RobustQA suite (https://aclanthology.org/2023.findings-acl.263.pdf):
>
> | model_name                                   |   RobustQA_Lifestyle |   RobustQA_Writing |   RobustQA_Science |   RobustQA_Recreation |   RobustQA_Technology |   average |
> |:--------------------------------------------|---------------------:|-------------------:|-------------------:|----------------------:|----------------------:|---------:|
> | Mistral-Small-24B-Instruct-2501             |                0.866 |              0.869 |              0.833 |                 0.779 |                 0.894 |    0.848 |
> | Oscar-llama 24B                             |                0.858 |              0.878 |              0.802 |                 0.751 |                 0.885 |    0.835 |
>
> Here, Oscar-llama remains close to its no-compression backbone, with 83.5% LLMEval compared to 84.8 %
>
> - Second, we also tested a non-QA RAG using the summarisation task of CNN/daily_mail. Here we measure Rouge metrics, as a proxy for content summarization compared to the ground truth summary:
> Model | Rouge-1 | Rouge-2 | Rouge-L
> -- | -- | -- | --
> Mistral | 0.21 | 0.04 | 0.19
> OSCAR-llama | 0.20 | 0.04 | 0.17
> OSCAR-llama maintains respectable performance on this summarization task.
>
> > How does the LoRA-fine-tuned generator perform on standard non-RAG benchmarks ? Does fine-tuning lead to a degradation in general-purpose reasoning or language understanding?
>
> Regarding non-RAG task, we note that since OSCAR is built using adapters, it is straightforward to just deactivate them at inference. Therefore, what matters and what we thoroughly tested is the all-domain RAG performance. Note also that HotpotQA is a multi-hop dataset which requires some form of 'reasoning'.
>
> > The experiments are mainly conducted on Mistral-7B and Qwen2-7B, while larger models such as Mistral-24B are only compared against an uncompressed version. Moreover, the evaluation focuses on accuracy and FLOPs, omitting critical system-level metrics such as end-to-end latency and throughput. Although Table 6 reports related values, it lacks direct comparisons to baseline methods.
>
> From our perspective, the number of floating point operations is the best efficiency metric that can be reported in our case. First, it does correlate to all the latency measurements (in ms) that we provide in Appendix Table 6. Second, the latency depends on a lot of parameters: batch size, hardware choice, document lengths etc… We believe our standardized measurements provide the best view. Finally, most of OSCAR latency improvements come from the generation part, which is directly comparable between models (all of them use the same generator LLM).
>
> > The paper claims that query-dependent embeddings preserve task-relevant semantics but provides limited theoretical analysis or formal justification to support this claim.
>
> First, query-aware compression enables to discard useless information at compression time and thus increases semantic relevance (as shown in https://arxiv.org/pdf/2407.15504). Second, the ablations in Table 2 directly show the performance impact of query-aware compression: it improves results by a few points, especially at high compression rates compared to query independent compression. Third, we show in appendix Figure 13, that the compressed embeddings do share more similarity (measured with cosine) with context which is relevant to the query, on a synthetic needle-in-a-haystack test. This figure is a direct visualization of the query-aware effect on the compression.
>
> > Could the authors report results when the generator is kept frozen during training to isolate the effect of compression?
>
> Yes here they are for OSCAR-N-Layers:
> | model_name                     |  ASQA | Hotpotqa |   NQ  | Triviaqa | POPQA | BIOASQ12B | Average |
> |--------------------------------|------:|---------:|------:|---------:|------:|----------:|--------:|
> | Mistral-7B-Instruct-v0.2       | 0.752 |    0.512 | 0.683 |    0.917 | 0.699 |     0.510 |  0.679  |
> | OSCAR-8-layers                 | 0.744 |    0.528 | 0.674 |    0.917 | 0.678 |     0.516 |  0.676  |
> | OSCAR-8-layers frozen decoder  | 0.692 |    0.470 | 0.638 |    0.893 | 0.629 |     0.459 |  0.630  |
> They show an 8% decrease in accuracy. We'll add this as an extra ablation in Table 2 in the revised paper.

---

### Official Review · Reviewer_6CpB · 2025-11-01

**Soundness:** 2
**Presentation:** 3
**Contribution:** 3
**Rating:** 6
**Confidence:** 3

**Summary:**

This paper propose a compression method for RAG, called OSCAR. This method includes two designed models; the first one is called compressor, which is used to convert document chunks as well as query into fixed-length embeddings. The second model is called generator, which leverages the query and the embeddings obtained from the compressor to generate an answer. Both compressor and generator can be trained in an end-to-end fashion, using the RAG results from a teacher model without compression as training signals. Experiment results show that on a series of QA datasets, the proposed method can achieve similar performance compared to no compression and slighter better inference cost.

**Strengths:**

1. The paper is well writen and easy to follow.
2. The proposed method combines the advantages of both hard-compression and soft-compression, allowing online query-aware compression while ensures a better compression rate. It is a pretty straightforward and intuitively nice idea.
3. The experiment results are conducted on a wide range of datasets, making the rseults look solid.

**Weaknesses:**

One possible concern is about the context length. This paper majorly conducts experiments on 128 token chunks, 5/10 chunks per query. This is one possible scenario, but other cases where longer context is applied is not tested in this paper. I would argue that results only under the settings from this paper is not comprehensive enough.

**Questions:**

Evidence that the method generalizes to longer context cases would be appreciated; does it maintain similar accuracy and end-to-end speed when k and document lengths increase?

---

> ### Author Response · Authors · 2025-11-19
>
> Thank you for your feedback, and for acknowledging the robust set of experiments we ran to illustrate OSCAR abilities.
>
> > Evidence that the method generalizes to longer context cases would be appreciated; does it maintain similar accuracy and end-to-end speed when k increases?
>
> Figure 5 shows OSCAR is generalizing as well as its underlying backbone to the number of documents k, top to k=50.
> Regarding inference speed, improvements are maintained for every value of k, and proportional to k. Additionally, batch sizes my be increased for OSCAR, enabling higher level of distribution.
>
> > Does it maintain similar accuracy when the document length increases ?
>
> Regarding document lengths, we would like to discuss a few points. First, chunking is a part of RAG pipelines: it allows the retrieval to identify specific documents which are of relevance in the document collection. Therefore, chunking at 128 is not a limitation per se. Also please note that our RAG results without compression are high and constitute a very strong RAG benchmark, further highlighting that this 128-token is not a strong limitation.
>
> Second, to gain insights into the robustness of OSCAR to longer documents, we perform new experiments where:
> - The top 10 retrieved documents are concatenated into 5 documents twice as long. This test pure ability to exploit “long” context (denoted **5x2 docs**)
> - The top 10 retrieved documents are concatenated with random documents to from 10 new documents twice as long (denoted **10 x (doc + noisy doc)**)
>
> | model_name   |    ASQA |   Hotpotqa |    NQ |   Triviaqa |   POPQA |   BIOASQ12B | Average |
> |:-------------------------------------------|--------:|-----------:|------:|-----------:|--------:|------------:|------------:|
> | Mistral-7B    |   0.752 |      0.512 | 0.683 |      0.917 |   0.699 |       0.51 | 0.679|
> | Mistral-7B 5x2 docs                   |   0.732 |      0.484 |  0.650 |  0.911 |   0.649 |     0.495  | 0.654 |
> | Mistral-7B 10 x (doc + noisy doc)      |   0.674 |      0.513 |  0.682   |  0.918 |   0.696 |     0.502     |  0.664|
> | OSCAR-llama  |  0.742 |      0.527 | 0.682 |      0.917 |   0.685 |       0.518 |   0.679
> | OSCAR-llama  5x2 docs      |  0.703  |       0.494 | 0.648 |   0.904 |   0.615 |     0.495 |   0.643 |
> | OSCAR-llama 10 x (doc + noisy doc)   |  0.715 |      0.503 | 0.651 |   0.909 |   0.648 |     0.510 | 0.656 |
>
> On the 5x2 docs experiments, OSCAR, although not trained on 256 token documents, suffers only a slight decrease in accuracy (-1%), but still reaches decent results.
>
> When the 10 documents are mixed with noisy documents (10x(doc + noisy)), OSCAR is about as strong as its Mistral-7B backbone (-0.7%). This underlines the abilities of OSCAR to handle longer context.
>
> We will update soon the paper with revisions also including comments from other reviewers.

---

### Official Review · Reviewer_4qhT · 2025-11-02

**Soundness:** 3
**Presentation:** 3
**Contribution:** 3
**Rating:** 6
**Confidence:** 3

**Summary:**

The paper introduces OSCAR (Online Soft Compression for RAG), which tries to "sit in the middle" between (i) online, query-aware hard pruning methods such as Provence and RECOMP, and (ii) offline, high-ratio soft compression methods such as PISCO / earlier context-embedding work. The key idea is to run, at inference time, a query-conditioned compressor LLM that takes (q, dᵢ, MEM tokens) and emits a small set of embedding tokens per retrieved document, which are then fed to the generator LLM instead of the original text. Two realizations are proposed: (1) OSCAR-N-Layers (reuse the first N layers of the generator) and (2) OSCAR-llama (a small 1B Llama with a learned projection). The whole pipeline is trained end-to-end by sequence-level distillation from an uncompressed RAG teacher (Mistral-7B) and can optionally output a reranking score from the same forward pass so that compression is "for free" On six RAG QA benchmarks and on several backbones (Mistral-7B, Qwen2-7B, Mistral-24B, Llama-1B) the authors report 2–5× FLOP savings with little or no accuracy loss relative to uncompressed RAG and better speed/quality than online hard pruning (Provence, RECOMP).

**Strengths:**

- Well-framed gap and plausible contribution. The paper correctly observes that current online methods are mostly text-level and top out around 2× compression (Provence, RECOMP) while soft methods (PISCO, xRAG/COCOM-style context embeddings) reach 16× but are practically offline and often query-independent. Positioning OSCAR as "query-dependent and online" is a nice conceptual sweet spot.

- Strong and unusually broad empirical section. The paper does not just show "one backbone, one dataset": it repeats the story on four backbones (including a 24B model), six RAG QA tasks, and also checks harder settings such as BM25-only retrieval, more retrieved docs (up to 50), different compressor depths, and different compressor families. That breadth makes the claim “this is not overfit to one setup” fairly convincing.

- Factoring compression and reranking in one pass is practically useful. Existing strong RAG pipelines already pay for a cross-encoder reranker; using (almost) the same computation to also produce compressed representations is exactly the kind of engineering win production RAG wants, and it is consistent with how Provence also couples pruning and reranking.

**Weaknesses:**

- Novelty is a bit overstated. The paper calls OSCAR the "first query-dependent online soft compression," but there is prior art that is close enough that it must be discussed carefully:

    - DODO already performs dynamic contextual compression for decoder-only LMs and can be used as a context compressor at inference. It also reduces the number of hidden states and is explicitly meant to lower long-context cost.

    - PISCO (2025) explicitly targets 16× compression for RAG and is trained by sequence-level distillation; the main difference is "offline vs. online," but OSCAR’s evaluation makes PISCO pay an online cost, which weakens the fairness of the comparison.

    - Provence / RECOMP already couple pruning with reranking or selective augmentation; OSCAR's "free compression + rerank" is in the same design space.

- The paper needs to spell out why these do not qualify as "online soft, query-dependent" in the authors' definition.

- Comparison protocol slightly favors OSCAR. Counting PISCO’s compression as if it had to be done online makes OSCAR look 3–5× faster, but the whole point of PISCO/xRAG/COCOM-like approaches is that you pre-compute or at least pre-compress. Likewise, Provence and RECOMP are mostly LLM-agnostic and don't require per-backbone re-training, while OSCAR has to be retrained for every generator (the authors themselves retrain for Mistral-7B, Qwen2-7B, Mistral-24B, Llama-1B). Deployment cost and model storage are not reported, so the advantages are not end-to-end in a systems sense.

- Missing or underdeveloped baselines / metrics. The related-work section knows about FINCH, TurboRAG / block-attention RAG, and long-context KV-cache tricks but they are not in the experiments, even though some of them also run online and trade KV size for speed. We also never see actual wall-clock latency or QPS-under-load numbers, only FLOPs on synthetic inputs, so it’s hard to tell whether "OSCAR compressor + generator" < "Provence + generator" on a real A100/H100 with retrieval in front. And everything is single-domain QA with 128-token chunks; real RAG often has heterogeneous, noisy, and much longer pieces.

**Questions:**

- Scope of "online." In a production RAG service that already runs a DeBERTa-v3 reranker, what is the measured latency (ms) of OSCAR-llama / OSCAR-8-Layers for 10×128-token docs on an A100, and how does that compare to that reranker? Please report wall-clock, not only FLOPs.

- Reusability across queries. Because OSCAR is query-conditioned, you can't pre-compress a corpus once. Did you try a two-stage variant (cheap query-agnostic soft compression at index time + light query adapter at run time)? If not, what prevents it?

- Positioning vs. DODO / TurboRAG / KV-cache compression. What concrete capability do those methods lack that forces you to call OSCAR "the first" online soft compression? In particular, DODO can also run at inference and reduce hidden states.

- Sensitivity to retrieval quality. You show BM25-only experiments, but only aggregated. Can you provide per-dataset breakdowns and failure cases (e.g., when none of the top-k docs is actually relevant), and how compression interacts with that?

- Teacher choice. You always distill from Mistral-7B. If the teacher is smaller or of lower RAG quality, do you still beat Provence / RECOMP?

---

> ### Author Response · Authors · 2025-11-19
>
> Thanks for your feedback. We share your point of view that oscar hits a sweet-spot 'online + query-dependent' enabling good performances and practical use in real applications. We acknowledge your comments and address them below.
>
> > DODO already performs dynamic contextual compression for decoder-only LMs and can be used as a context compressor at inference. [...]
>
> DODO is definitely a relevant reference which we will add in the related works of the paper. DODO however does not propose a mechanism to build document-wise embeddings and use them for generation. It can be used directly as a 'long-context' method on the full RAG context, but this does not lead to satisfying performances (as shown in PISCO, Table 2 https://arxiv.org/abs/2501.16075)
>
> > [...] OSCAR’s evaluation makes PISCO pay an online cost, which weakens the fairness of the comparison.
>
> Thank you for raising this point. We did provide the separate measurements for the compression/generation, and the generation-only cost of PISCO in Table 1: they do show that the fastest-at-inference solution is always PISCO. However, OSCAR operates without precomputing and storing any of the compressed representations. This makes it straightforward to handle very large collections, as well as collections that evolve over time—as is often the case in real applications—without the need to continually refresh stored representations. Depending on system load and document distribution, it can also avoid substantial offline compute costs that PISCO requires. Finally, OSCAR has the additional advantage of being more accurate since the compression is query-dependent (see Table 2).
>
> > OSCAR's "free compression + rerank" is in the same design space [as Provence]
>
> Yes, we explicitly acknowledge Provence for the simultaneous reranking idea and do not claim this idea as a novel contribution. Our contribution lies in **making it work in the context of soft compression**, which is significantly more challenging. In Provence, the compression and reranker are trained jointly in a fully supervised setting, whereas in OSCAR the text supervision comes from the generation signal (backpropagated through the generator) while the reranking supervision comes from labels. This makes the joint training—especially balancing generation quality with reranking quality—considerably harder. From this perspective, our reranking results demonstrate that simultaneous reranking is feasible for soft compression, and the hyperparameters we provide (notably λ) offer practical value for reproducing and extending this result.
>
> > Comparison protocol slightly favors OSCAR. Counting PISCO’s compression as if it had to be done online makes OSCAR look 3–5× faster, but the whole point of PISCO/xRAG/COCOM-like approaches is that you pre-compute or at least pre-compress. Likewise, Provence and RECOMP are mostly LLM-agnostic and don't require per-backbone re-training, while OSCAR has to be retrained for every generator (the authors themselves retrain for Mistral-7B, Qwen2-7B, Mistral-24B, Llama-1B). Deployment cost and model storage are not reported, so the advantages are not end-to-end in a systems sense.
>
> The online pipeline studied in our paper has storage benefits, flexibility advantages and is also more accurate. Pre-compressing is a different setup which holds more inference-time latency improvements but has storage/compute/flexibility/accurate downsides.
> We do need to retrain for each backbone, but training time is limited (4 A100-gpu-days for the 24B model) and is done with LoRA so additional storage is limited (1-2% of the model total parameters). In fact, we measure that OSCAR training time is lower than the time it takes to pre-compress 5M documents with PISCO. Also, PISCO storage requires 64kB per document which already amounts to 320GB for 5M docs.
>
> > [...] The related-work section knows about FINCH, TurboRAG / block-attention RAG, and long-context KV-cache tricks but they are not in the experiments,
>
> KV-cache compression methods are not specific to RAG and in general do not offer document-level compression but context-level. Such methods may be used on top of OSCAR at generation and thus are orthogonal in the way they improve efficiency.  That said, we did run experiments with such methods (specifically https://arxiv.org/pdf/2510.00636)  and have several takeaways. First, latency gains are obtained only for large contexts (>~4k tokens) as the KV compression itself has an added cost. Second, we obtained the following accuracies on ASQA with Mistral-7B:
> | Model                            | ASQA  |
> |----------------------------------|-------|
> | Chunk-Press, compr 4x      | 0.681 |
> | Chunk Press, compr 2x       | 0.691 |
> | Chunk-Press, compr 1.3x     | 0.659 |
> | No compression        | 0.739 |
> which show a strong performance drop even at moderate compression rates.
>
> [CONTINUED IN NEXT POST]

---

> > ### Author Response · Authors · 2025-11-19
> >
> > [CONTINUED PREVIOUS POST]
> >
> > > We also never see actual wall-clock latency or QPS-under-load numbers, only FLOPs on synthetic inputs, so it’s hard to tell whether "OSCAR compressor + generator" < "Provence + generator" on a real A100/H100 with retrieval in front.
> >
> > Wall-clock latency are provided in the Appendix (Table 6). They are in line with FLOPS measurements. On Table 6, we also provide maximum peak memory usage: OSCAR uses much less memory due to the shortened context. This means that if deployed in production system, batch sizes could be increased, further improving the latency and economics of the pipeline. Note that we systematically exclude the retrieval cost for all models as it is identical across all models and around ~1-5ms for 10M collections: negligible compared to the generation cost.
> >
> > > Scope of "online." In a production RAG service that already runs a DeBERTa-v3 reranker, what is the measured latency (ms) of OSCAR-llama / OSCAR-8-Layers for 10×128-token docs on an A100, and how does that compare to that reranker? Please report wall-clock, not only FLOPs.
> >
> > As mentionned above, see Appendix Table 6. Latency gains correlate with measures FLOPS. DeBERTa-v3 performance can be read from Provence which uses that backbone.
> >
> > > Reusability across queries. Because OSCAR is query-conditioned, you can't pre-compress a corpus once. Did you try a two-stage variant (cheap query-agnostic soft compression at index time + light query adapter at run time)? If not, what prevents it?
> >
> > That’s an interesting question. PISCO already does expensive query-agnostic soft compression with no further compression and suffers from a performance drop compared to OSCAR. Therefore, we do not expect any gains from the proposed pipeline and a ‘cheaper’ agnostic compression. Second, this proposed setup would have the downside of offline compression (computing, storing and maintaining an index) AND the downside of online compression (smaller latency, albeit much smaller compression cost in this case). So nothing prevents it, but it’s not a valuable setup.
> >
> > > What concrete capability do those methods [PROVENCE,DODO,KV caches] lack that forces you to call OSCAR "the first" online soft compression? In particular, DODO can also run at inference and reduce hidden states.
> >
> > It is true that in some sense all of these methods do perform an online soft compression. OSCAR's is document-wise, and more importantly, leads to nearly **lossless performances**. We'il replace 'the first' with 'a novel' nonetheless in the revised paper.
> >
> > > You show BM25-only experiments, but only aggregated. Can you provide per-dataset breakdowns and failure cases (e.g., when none of the top-k docs is actually relevant), and how compression interacts with that?
> >
> > These results are provided on Figure 8 in the appendix but there is an error on the labels: each individual subplot is a separate dataset (from left to right, top to bottom: ASQA, HotpotQA, NQ, TriviaQA, POPQA, BioASQ). We will correct that figure in the revised version.
> >
> > The drop in performance is slightly higher for OSCAR than for the no-compression baseline. This suggests that OSCAR may be a bit more sensitive to noisy retrieval. Future work may try to address this e.g. by including negative documents at train time as in RAFT (https://arxiv.org/abs/2403.10131)
> >
> > > Teacher choice. You always distill from Mistral-7B. If the teacher is smaller or of lower RAG quality, do you still beat Provence / RECOMP?
> >
> > We did not study other teachers for OSCAR. PISCO paper provides an ablation on this (see https://arxiv.org/pdf/2501.16075 Figure 7) and we relied on their results.

---

### Author Response · Authors · 2025-11-27

Dear Reviewers,

Thank you again for your time and thoughtful feedback—we truly appreciate your efforts.

We hope that our rebuttal successfully addressed your questions and provided clearer evidence of the effectiveness of our approach. If there are still points that remain unclear, we would be grateful if you could share any updates or additional comments soon, as this would help ensure that we have sufficient time to respond before the discussion phase concludes.

---

### Author Response · Authors · 2025-12-03

Dear AC and Reviewers,

Thank you very much for your detailed reviews. We appreciate the time and care you dedicated to evaluating our work.

We are grateful for the feedback regarding OSCAR’s core contributions. Several reviewers highlighted the clear motivation of bridging hard and soft compression, the practical value of fully online and query-dependent compression, the strong empirical results across multiple backbones and datasets, and the usefulness of integrating compression and reranking in a single pass. We are pleased that the intended strengths of OSCAR were well understood and appreciated.

During the discussion phase, we provided additional experiments and clarifications that further show the robustness and value of the approach. It will be fully reflected in the revised paper. These include:
- Expanded comparisons and positioning within related online and soft compression approaches, clarifying OSCAR’s contribution as a novel online, query-dependent soft compression method.
- Additional system-level and latency measurements, confirming that the reported FLOPs savings translate into practical efficiency gains and reduced memory usage.
- New experiments on longer contexts, cross-domain retrieval, and non-QA RAG tasks, showing that OSCAR remains aligned with its backbone across settings while maintaining strong efficiency–accuracy tradeoffs.
- Additional ablations on training strategies and compressor/generator configurations, further illustrating the stability and generality of the approach.

We believe these updates help present a clearer and more complete picture of OSCAR’s capabilities and practical value for efficient RAG systems.

Thank you again for your engagement.

---

### Meta-Review · Area_Chair_7eoP · 2025-12-16

**Summary:**

Initial scores are 8, 6, 6, 6. Reviewers acknowledged the well-designed method and comprehensive experiments showing 2-5× speedup with minimal accuracy loss, but raised concerns about: (1) overstated novelty claims ("first online soft compression") given prior work (DODO, PISCO online mode), (2) per-backbone retraining cost, (3) limited evaluation on longer contexts (primarily 128-token chunks), (4) potential generalization issues from fine-tuning the generator, and (5) missing system-level metrics beyond FLOPs in main results.

**Reviewer Concerns:**

**Addressed:**
- **Longer context evaluation (6CpB):** Authors provided experiments with 256-token docs (5×2 setup) showing 1% accuracy drop, and 10×(doc+noisy) experiments demonstrating robustness to longer contexts.
- **Cross-domain generalization (j7H9):** Authors added RobustQA evaluations (83.5% vs 84.8% baseline) and CNN/DailyMail summarization results showing respectable transfer.
- **Generator fine-tuning impact (j7H9):** Ablation with frozen decoder shows 8% drop, justifying LoRA fine-tuning. Authors note LoRA enables easy deactivation for non-RAG tasks.
- **System-level metrics (4qhT, j7H9):** Authors point to Appendix Table 6 with wall-clock latency measurements correlating with FLOPs.
- **BioASQ underperformance (kf7i):** Authors provided additional RobustQA results demonstrating competitive cross-domain performance.

**Outstanding:**
- **Novelty positioning (4qhT):** Authors conceded to change "first" to "novel" but DODO/PISCO comparison protocol concerns remain. DODO applicability to document-wise RAG not fully resolved; PISCO comparison fairness (online cost attribution vs. intended offline use) remains debatable.
- **Per-backbone training cost (kf7i, j7H9):** Authors defend 4 GPU-days as acceptable and note performance follows backbone, but operational cost for switching LLMs remains a practical limitation versus training-free methods.
- **FiD-light/TurboRAG comparisons (kf7i):** Authors mention FiD-light has low compression (50 emb/token) and KV-cache methods require >4k tokens, but direct empirical comparisons with these online methods missing.

**Reviewer Scores:**

- **Reviewer 4qhT (initial: 6):** Would likely remain at 6. Core novelty concerns about positioning vs. DODO/PISCO remain unresolved despite authors' clarifications. The rebuttal provides useful context but doesn't fully address the "first online soft compression" overclaim.

- **Reviewer 6CpB (initial: 6):** May increase to 8. Primary concern about longer context generalization was well-addressed with new 256-token and noisy document experiments. Results demonstrate acceptable robustness.

- **Reviewer j7H9 (initial: 6):** May increase to 8. Comprehensive response addressing cross-domain evaluation (RobustQA, CNN/DM), frozen decoder ablation, and non-RAG task handling through LoRA deactivation. Generalization concerns substantially mitigated.

- **Reviewer kf7i (initial: 8):** Would likely remain at 8. Mostly positive initial assessment; rebuttal addressed BioASQ questions and provided reasoning for baseline choices. Missing FiD-light comparison is minor.

---

### Decision · Program_Chairs · 2026-01-26

Accept (Poster)